# Remediating Potentially Toxic Metal and Organic Co-Contamination of Soil by Combining In Situ Solidification/Stabilization and Chemical Oxidation: Efficacy, Mechanism, and Evaluation

**DOI:** 10.3390/ijerph15112595

**Published:** 2018-11-20

**Authors:** Yan Ma, Zhenhai Liu, Yanqiu Xu, Shengkun Zhou, Yi Wu, Jin Wang, Zhanbin Huang, Yi Shi

**Affiliations:** 1School of Chemical and Environmental Engineering, China University of Mining and Technology (Beijing), Beijing 100083, China; hiram0823@163.com (Z.L.); xuyq15871041327@163.com (Y.X.); 18763823855@163.com (S.Z.); 13121966311@163.com (Y.W.); 17810286705@163.com (J.W.); zbhuang2003@163.com (Z.H.); 2State Key Laboratory of Environmental Criteria and Risk Assessment, Chinese Research Academy of Environmental Science, Beijing 100012, China

**Keywords:** aniline, cadmium, combined remediation, ISCO, ISS, persulfate

## Abstract

Most soil remediation studies investigated single contaminants or multiple contaminants of the same type. However, in field conditions, soils are often contaminated with potentially both toxic metals and organic pollutants, posing a serious technical challenge. Here, batch experiments were conducted to evaluate the performance of combining in situ solidification/stabilization (ISS) and in situ chemical oxidation (ISCO) for the simultaneous removal of aniline (1000 mg/kg) and Cd (10 mg/kg). All four tested ISS amendments, especially quick lime and Portland cement, promoted in situ chemical oxidation with activated persulfate in contaminated soil. Combined ISS/ISCO remediation effectively removed aniline and reduced the bioavailable Cd content at optimal initial persulfate and ISS amendment concentrations of 1.08 mol/kg and 30 wt% with a seven-day curing time, and significantly reduced leaching. Persulfate inhibited the reduction of the bioavailable Cd content, and ISS amendment with persulfate did not synergistically remediate Cd in co-contaminated soil. Strong alkalinity and high temperature were the main mechanisms driving rapid pollutant removal and immobilization. The reaction of CaO with water released heat, and Ca(OH)_2_ formation increased the pH. The relative contributions of heat vs. alkaline activation, as well as the contaminant removal efficiency, increased with ISS amendment CaO content. Combined treatment altered the soil physicochemical properties, and significantly increased Ca and S contents. Activated persulfate-related reactions did not negatively impact unconfined compressive strength and hydraulic conductivity. This work improves the selection of persulfate activation methods for the treatment of soils co-contaminated with both potentially toxic metals and organic pollutants.

## 1. Introduction

Rapid development of industrial and mineral industries can cause serious contamination of soil by multiple pollutants [1,2]. In China, co-contamination of soil has become a major feature of contaminated sites, typically as a combination of organic substances and potentially toxic metals (for typical pollutants in China, see Appendix A) [3]. For instance, aniline, a potential endocrine disruptor that also induces immunotoxicity, neurovirulence, and other effects, is an organic compound used widely as a raw material in the national defense, pesticide, pharmaceutical processing, plastic product, and printing and dyeing industries [4]. Moreover, industrial waste often contains large amounts of potentially toxic metals; for example, the disassembly and disposal of electronic waste inevitably results in metal pollution. Among potentially toxic metals, Cd pollution is common, and Cd can enter the human body via the digestive and respiratory systems where it can accumulate, threatening human health [5]. Co-contaminated soils pose a greater hazard than soils contaminated with single compounds, but are more difficult to remediate [6]. As such, there is increasing attention being paid to the development of remediation technologies that target co-contaminated soils in recent years.

The main remediation methods applied to soil contaminated with potentially both toxic metals and organic compounds include physicochemical remediation and bioremediation [7,8]. Of these, physicochemical remediation can be used to treat soil and groundwater contaminated with high concentrations of potentially toxic metals and organic matter [9,10,11,12,13]. For instance, Song et al. [14] conducted batch experiments to evaluate the performance of saponin (a plant-derived biosurfactant) for the simultaneous removal of polycyclic aromatic hydrocarbons (PAHs) and Cd from co-contaminated soils, and suggested that saponin could remove heavy metals (55.8%) and PAHs (>90%) from co-contaminated soil. Meanwhile, Sui [15] synthesized a mixture of submicron Fe oxides on the basis of sludge hydrolysate, and found a tetrabromobisphenol A removal efficiency of 53.5% and Cd immobilization efficiency of 42.96% after a 15-day reaction. In addition, Wang et al. [16] conducted a study of the simultaneous removal of phenanthrene and Pb from artificially contaminated soils using glycine-beta-cyclodextrin. However, to our knowledge, no studies of simultaneous aniline–Cd remediation were reported.

In situ solidification/stabilization (ISS) is used as a stand-alone method to remediate and redevelop manufactured gas plants, brownfields, and Superfund sites. ISS involves mixing cementitious reagents with contaminated soil to reduce the physically and/or chemically available solution and the bioavailability and leachability of contaminants, while improving the soil characteristics [17,18]. Reagents containing CaO are widely used in the immobilization of heavy metal-contaminated soil, including Cd [19,20]. However, many problems exist regarding the use of solidification/stabilization to remediate organic compound-contaminated soil, mainly because organic compounds do not undergo chemical reaction during the solidification/stabilization process with ISS amendments. Since the immobilization efficiency mainly depends on the physical/physicochemical stabilization process, pollutants can readily leach from solidified matrices, posing risks to the environment. For example, Sora et al. [21] suggested that more than 75% of 2-chloroaniline could leach from a cement solidification matrix. In addition, organic compounds can influence the stability of solidified matrices, impeding the creation of effective curing technologies [22,23].

More recently, sulfate radical (SO_4_^·−^)-based activation oxidation technologies gained attention in remediation applications. SO_4_^·−^ can be produced from activated persulfate (PS) and peroxymonosulfate by heat, bases, transit metals, ultraviolet light, and sonolysis. In particular, heat activation requires temperatures above 30 °C, and alkaline activation takes place at a pH above 10.5 [24]. SO_4_^·−^ has a relatively high standard oxidation–reduction potential (ORP) of 2.6 V and is capable of degrading aniline, PAHs, and many other contaminants [25]. Moreover, SO_4_^·−^ can react with water to form the hydroxyl radical, which has an ORP of 2.8 V and also oxidizes a wide variety of organic pollutants, including aniline [26]. Although in situ chemical oxidation (ISCO) degrades organic contaminants effectively, particularly highly soluble contaminants [27], high oxidant doses and multiple injections are often required, and the residual contaminants are not prevented from leaching.

Combined treatments to overcome the limitations of ISCO were investigated to improve remediation efficiencies [28]. For instance, chemical oxidants are generally more expensive than ISS amendments and, although ISS leaves most contaminants in situ, it reduces their leachability potential and improves soil properties. As such, the limitations of both ISCO and ISS may be minimized by combining these two technologies. In addition, CaO-containing ISS reagents can react with water to form alkaline environments and release heat, which could create favorable conditions for PS activation.

In this study, treatment of aniline–Cd co-contaminated soil using combined ISS/ISCO was conducted. The main objectives were to (1) develop an efficient technique for co-contaminant remediation by assessing the aniline removal efficiency and bioavailable Cd content, (2) explore the mechanism of this process and the role of CaO, and (3) evaluate the treatment effect on soil properties and long-term stability of the solidification/stabilization products. The results obtained in this study are expected to provide an efficient method for the treatment of potentially toxic metal/organic compound co-contaminated soil in future engineering applications.

## 2. Materials and Methods

### 2.1. Chemicals and Materials

#### 2.1.1. Chemicals

Aniline (99%) was purchased from Sigma-Aldrich (St. Louis, MO, USA). All solvents (methanol, methyl tert-butyl ether, and acetone) were of HPLC grade (J.T. Baker Inc., Phillipsburg, NJ, USA). Sodium persulfate (Na_2_S_2_O_8_), ammonium thiosulfate (Na_2_O_2_S_3_), cadmium tetrahydrate (Cd(NO_3_)_2_·4H_2_O), ethylenediaminetetraacetic acid (EDTA-2Na), diethylenetriaminepentaacetic acid (DPTA), and triethanolamine (TEA) were purchased from Sinopharm (Beijing, China). Ultrapure MilliQ water (EMD Millipore, Billerica, MA, USA) was used in all experiments. All chemical reagents and organic solvents were of at least analytical grade and were used as received without further purification.

#### 2.1.2. Test Soil Sampling and Processing

A clean soil sample was collected from surface soil (0–20 cm) of farmland in Tongzhou District, Beijing, China. Soil drilling was used as a means to collect soil. The surface debris at the sampling point was removed before the soil was collected, and the drill was inserted vertically into the soil to a depth of 0–20 cm (marked on the drill with a scale). The soil was sieved to remove particles larger than 0.5 cm in diameter, to remove stones and plant debris [24], and was homogenized. The physicochemical properties of the soil samples are shown in Appendix A. Cd-contaminated soil was prepared by mixing soil with a Cd(NO_3_)_2_·4H_2_O solution. Aniline-contaminated soil was prepared by dissolving an appropriate quantity of aniline in 100 mL acetone and slowly adding a known weight of soil with continuous mixing. Then, this slurry was shaken at 250 rpm for 30 min in a 25 °C constant temperature air bath shaker to promote homogenization of aniline in the soil and the solvent was allowed to evaporate slowly. The dry contaminated soil was transferred to a sealed container for aging. Co-contaminated soil containing Cd and aniline was prepared in two sequential steps: clean soil was first spiked with Cd and then with aniline, as described above [14,28]. The contaminated soil had a final concentration of 10 mg/kg Cd and 1000 mg/kg aniline in both the single- and co-contaminated soils. These concentrations were derived from previous studies that found high concentrations of heavy metals and organic pollutants in co-contaminated soil [5,29].

#### 2.1.3. In situ Solidification/Stabilization (ISS) Amendments

Four ISS amendments with different properties, including varying CaO contents, were applied in this study (Table 1). The amendments included quick lime (QL), Portland cement (PC), blast-furnace slag (BFS), and fly ash (FA).

### 2.2. Aniline Oxidation by Persulfate (PS)

Aniline oxidation was performed based on previously described methods [32,33]. Batch degradation studies were conducted in 100-mL stoppered glass conical flasks containing 20 g of soil. Then, 20 mL of PS solution with an initial concentration of 53.64 mmol/L, 107.52 mmol/L, 215.06 mmol/L, 537.64 mmol/L, 1.08 mol/L, or 2.15 mol/L (with molar ratios to aniline of 5, 10, 20, 50, 100, and 200) were added to the reactors. All reactors were kept in a constant-temperature oscillator at a temperature of 25 °C and rotation speed of 150 rpm. The analytical samples were removed at different reaction times (10 min, 30 min, 60 min, 90 min, 150 min, and 210 min), and the appropriate amount of Na_2_S_2_O_3_ solution (2 mol/L) was added to carry out the quenching reaction. Each analytical sample was centrifuged at 3000 rpm for 30 min. The supernatant was placed in a separatory funnel, and extracted with methyl tert-butyl ether (MTBE) after centrifugation. The centrifuged soil was dried in a freeze-dryer for 12 h, and then extracted with MTBE. The extracts were used to measure the aniline concentration. The effects of reaction temperature (10 °C, 20 °C, 30 °C, 40 °C, and 50 °C) and pH (3, 5, 7, 9, and 11) on aniline degradation and total organic carbon (TOC) removal in soil were investigated based on the above reactor conditions. All experiments were performed in triplicate, and the standard deviations were calculated.

### 2.3. Persulfate (PS) Oxidation and Soil Solidification/Stabilization

The reactors were 1-L closed cylindrical plexiglass tanks with two ports. One port housed the shaft of a propeller attached to a mixer. Mixing at 200 rpm was performed to homogenize the reactor contents and soil samples. The other port was used to insert pH and temperature probes. Each reactor was charged with 1 kg of contaminated soil (dry weight). The water content of the homogenized soil was 10%. In addition to the retained soil water, ultrapure water was added at a volume of 30% (v/w water/dry soil). After adding the ISS amendments to the soil, PS was immediately added into the reactor (as a solid powder) and the oxidation process began. Appendix A lists all of the reaction scenarios tested, the name for each used throughout the paper, and the concentrations of PS, ISS, and Ca(OH)_2_ added. After 3 h of reaction, the mixture was poured into molds (7 cm × 7 cm × 7 cm), filling them half way. The half-filled molds were compacted using a vibrating table for approximately 10 s, and then completely filled. The molds were further compacted and the excess mixture was scrapped off to obtain a flat and smooth surface. The molds were sealed in plastic bags for seven days in a chamber with a relative humidity of 95 ± 5% and a temperature of 20 ± 2 °C. After curing, the samples were demolded. One portion of each sample was used for measurement, and the other was resealed in a plastic bag and transferred back into the humidity chamber for 28 days of curing for the unconfined compressive strength (UCS) and hydraulic conductivity (K) analyses. Each treatment was performed in triplicate.

### 2.4. Analytical Methods

The analyses of soil organic matter (SOM) and cation exchange capacity (CEC) were mainly based on the method recommended by Zhang et al. [34]. The pH, electrical conductivity (EC), and ORP were tested using a water quality instrument (YSI, Yellow Springs, Ohio, US). Temperature was measured using a thermometer. The elements Ca and S were measured using an iCAP6300 inductively coupled plasma spectrometer (Thermo Fisher Scientific, Waltham, MA, USA). The aniline concentration was measured with gas chromatography/mass spectrometry (GC, CP-3800; MS, Saturn Ion Trap 2200; Varian, PaloAlto, CA, USA). The TOC analysis was conducted using a carbon analyzer (VCPN model; Shimadzu, Kyoto, Japan). The synthetic precipitation leaching procedure (SPLP) tests were performed on 50-g soil samples following United States Environmental Protection Agency (US EPA) Method 1312 using an extraction solution with a pH of 4.2 and a liquid-to-solid ratio of 20:1. The UCS test was conducted following ASTM D4219 (2009). X-ray diffraction (XRD) analysis was conducted using a diffractometer (D/max-3B; Ragaku, Tokyo, Japan) with Cu-K_α_ radiation to identify the phase formed after solidification/stabilization. Microstructure examination was performed using scanning electron microscopy (SEM; Zeiss Merlin, Jena, Germany).

The bioavailable Cd content in this study was characterized with a one-step extraction method that yielded CaCl_2_-Cd, EDTA-Cd, and DPTA-Cd [35]. CaCl_2_-Cd characterizes the mobility of Cd and has wide applicability to different soils, while EDTA-Cd and DPTA-Cd are mainly used to characterize the bioavailable fraction [36,37], and their comprehensive extraction can accurately characterize the bioavailability and mobility of Cd in soil. The specific extraction conditions of CaCl_2_-Cd, EDTA-Cd, and DPTA-Cd are shown in Table 2.

## 3. Results and Discussion

### 3.1. Oxidative Degradation of Aniline in Soil by PS

The concentration of PS has a critical role in activated PS oxidation of contaminants, because PS concentration can directly influence the equilibrium concentration of SO_4_^·−^. The effects of the initial concentration of PS on the oxidative degradation of aniline and TOC removal from soil are shown in Figure 1. Increasing the PS concentration within the range investigated (i.e., 53.76 mM to 2.15 M) effectively promoted the oxidative degradation of aniline. The degradation efficiency of aniline in soil was substantially removed within 10 min, and the degradation efficiency increased slightly at PS concentrations above 537.64 mM. This was inconsistent with the results obtained in aqueous solution, probably because aniline exhibits stronger migration in soil and possibly because greater ion exchange and redox conditions might occur in soil. The degradation and TOC removal rates reached 96.79% and 85.76%, respectively, with 1.08 M PS. After amendment with 2.15 M PS, the degradation and TOC removal rates further increased by only 1.01% and 1.69%, respectively. These results indicated that excess PS caused lower increases in aniline degradation and TOC removal, possibly because excess SO_4_^2−^ generated by PS hindered further reaction. Liang et al. [38] assessed the effects of PS treatment on organic polluted sand and silty soil, and found that the PS caused slow and mild oxidation of soil organic matter, resulting in a less-broken soil structure. In addition, the number of soil bacterial colonies increased after the reaction in PS-treated soil. Overall, our results suggest that PS has little effect on soil properties compared to other oxidants, and the treatment effect is more favorable.

Increasing the reaction temperature can effectively promote the decomposition of PS, thereby facilitating the formation of SO_4_^·−^. As a result, the aniline degradation rate increased with increasing temperature in the heat-activated PS system. Correspondingly, the degradation efficiency of aniline increased from 49.98% to 95.52% after 10 min of reaction and the TOC removal rate increased from 27.57% to 85.24% after 210 min of reaction as the temperature increased from 10 °C to 50 °C (Appendix A). The pH has a complex role in activated PS oxidation of contaminants. At low pH, PS can undergo acid-catalyzed decomposition, which depletes PS through non-radical pathways without producing SO_4_^·−^. At high pH, SO_4_^·−^ can be transformed into OH· via reaction with hydroxyl ions [28]. The effects of pH on the aniline degradation and TOC removal in soil are presented in Appendix A. The degradation efficiency of aniline followed the order pH 9 > pH 11 ≈ pH 7 > pH 5 > pH 3. The degradation efficiency of aniline in the soil environment with a pH of 5~11 was markedly higher than that of strongly acidic (pH = 3) soil, possibly because the −NH_2_ group in the aniline structure strengthened the adsorption of aniline onto soil in the acidic environment. Comparing the degradation rate of aniline in soil and water under the same pH condition, the removal effect in soil was much higher, possibly due to the improved soil mitigation effect on the change of pH and maintenance of the relative stability of the reaction environment, thereby promoting chemical degradation of aniline by PS. These results suggest that the oxidative degradation of aniline and removal of TOC in soil can adapt to a wider pH range.

### 3.2. Solidification/Stabilization of Cd-Contaminated Soil

The distribution, mobility, and bioavailability of potentially toxic metals in the environment depend not only on their total concentration, but also on the associated solid phase to which they are bound [39]. Figure 2 shows the bioavailable Cd content in contaminated soil, following the three extraction methods, after different treatments. The EDTA-Cd and DPTA-Cd concentrations were significantly reduced with the addition of ISS amendments after seven days of curing, while the CaCl_2_-Cd content decreased slightly, albeit non-significantly. The results indicate that lime-based agents can reduce the mobility of Cd in soil, reducing the environmental risk by minimizing bioaccumulation through the food chain in plants, animals, and humans. The bioavailability and, thus, plant absorption of Cd in soil was negatively correlated with pH. The addition of ISS amendments increased the pH of soil, promoting the formation of Cd(OH)_2_, CdCO_3_, and other precipitates and reducing the mobility of Cd.

Overall, the effect of the four ISS treatments on the solidification/stabilization of Cd followed the order QL > PC > BFS > FA. Meanwhile, there were minimal differences in the bioavailable Cd contents based on the three extraction methods. This finding suggested that the ISS amendments had significant effects on reducing the bioavailable Cd content in contaminated soil. In addition, the extractability of the three different extractants for Cd followed the order EDTA-Cd > DPTA-Cd > CaCl_2_-Cd, consistent with the sequential extraction capabilities of the three extractants. CaCl_2_ mainly extracts exchangeable heavy metals, while DPTA and EDTA can extract organic-bound heavy metals other than exchangeable heavy metals, and EDTA has a stronger complexing ability than DPTA [36]. These results indicated that the bioavailability of Cd in contaminated soil was greater than its mobility, and that organic-bound components accounted for the majority of the bioavailable content.

### 3.3. Optimization of Solidification/Stabilization Amendments and Dosage

#### 3.3.1. Selection of ISS Amendments

Sulfate radical-advanced oxidation effectively degraded aniline in soil, and ISS amendment significantly decreased the bioavailable content of Cd. Moreover, the addition of the four ISS amendments to the soil increased the temperature and pH of the medium, representing favorable conditions for the PS activation. Furthermore, while reducing the bioavailability of Cd in the co-contaminated soil, the residual aniline after chemical oxidation could be solidified via ISS amendment. Therefore, we hypothesized that the two methods, ISCO and ISS, could be applied in tandem to achieve synergistic effects. Cassidy et al. [27] recently documented that adding 50% PC and 50% hydrated lime could activate PS in contaminated soil, which chemically oxidized a significant portion of the contaminants.

To test this hypothesis, the aniline removal efficiency and bioavailable Cd content after combined treatment were determined in co-contaminated soil. As shown in Figure 3, the removal efficiency of aniline was only 48.21% with the addition of PS alone. However, the oxidative degradation of aniline significantly improved in the alkaline-activated PS system and heat-activated PS system, reaching 93.74% and 94.37%, respectively, of which the heat-activated PS system showed a slight advantage. Using the four ISS treatments alone, the degradation efficiency of aniline followed the order QL (63.34%) > PC (54.68%) > BFS (47.62%) > FA (46.42%), and revealed a low stabilization efficiency of aniline in co-contaminated soil (<90%), which could not meet environmental safety requirements. Similarly, Botta et al. [22] assessed 2-chloroaniline as a representative chlorinated aromatic amine pollutant, and found that it was difficult to immobilize in cement, consistent with the results of this study. These results indicate that ISS treatment alone cannot meet environmental safety requirements for the immobilization of aniline, presenting technical difficulties that must be addressed via combination with other remediation technologies to achieve suitable attenuation of aniline via solidification/stabilization.

When organic compounds are straightforwardly admixed with cement-based ISS amendments, they can affect the cement hydration kinetics via reactions that result in the formation of a protective film around the cement grain, hindrance of the formation of calcium hydroxide, and acceleration of the reaction via modification of the colloidal C-S-H (C = CaO, S = SiO, H = H_2_O) gel precipitated at very early stages around the cement grains. Such reactions may have been responsible for the observed low stabilization rate of aniline using PC. The combined ISS/ISCO treatments increased the contaminant removal efficiency compared to the ISS and ISCO treatments alone. Compared to the remediation effects of the four ISS amendments alone (treatments 4, 5, 6, and 7, Figure 3), the immobilization efficiency of aniline employing combined ISS/ISCO increased by 34.20%, 78.84%, 74.43%, and 68.12%, respectively. Treatment using QL and PC combined with PS could meet the evaluation criterion of an aniline immobilization rate greater than 90% [40]. This suggests that combined ISS/ISCO can solve the limitations of traditional ISS technology, which is not applicable to organic contaminants. In addition, combined ISS/ISCO remediation can not only reduce the hindrance effect of aniline on cement hydration kinetics, but also attenuate aniline concentrations, thereby effectively preventing the risk of secondary pollution caused by leaching of organic substances in the solidified matrix.

Figure 4 shows the effects of the tested treatments on the bioavailable Cd content in co-contaminated soil. The bioavailable Cd content after ISS treatment was significantly lower than treatment 1 (CK), treatment 2 (PS), and treatment 4 (Heat + PS), indicating that the application of ISS amendment regulated the bioactivity and reduced the mobility and environmental risk of Cd. The bioavailable Cd content in contaminated soil followed the order CaCl_2_-Cd < DPTA-Cd < ETDA-Cd, similar to that in the soil contaminated with Cd alone, and the organic-bound fraction remained the major component of bioavailable Cd.

Interestingly, the bioavailable Cd concentration after ISS treatment alone was much lower than that with combined remediation. Compared to the bioavailable Cd contents with the four ISS amendments alone (treatments 5, 6, 7, and 8, Figure 4), CaCl_2_-Cd increased by 35%, 28%, 126%, and 2997%, DPTA-Cd increased by 84%, 242%, 1213%, and 228%, and ETDA-Cd increased by 249%, 131%, 218%, and 19% after combined ISS/ISCO treatment. These results indicate that PS exhibits an inhibitory effect on the reduction of the bioavailable Cd content, and that PS and lime-based agents fail to have synergistic effects on Cd remediation in co-contaminated soil. This is plausibly attributed to the addition of SO_4_^2−^ and H^+^ during SO_4_^·−^-advanced oxidation processes, which lower the pH of the medium, thereby inhibiting the immobilization effect of ISS amendments on potentially toxic metals. Even though the combined ISS/ISCO treatments did not achieve synergistic remediation of Cd in co-contaminated soil, the bioavailable Cd content after combined remediation was much lower than the soil environmental quality standard (GB15618-1995).

#### 3.3.2. Optimal Dosage of Selected Amendments

The effects of the addition of different amounts of QL and PC on the bioavailable Cd content in co-contaminated soil are presented in Appendix A. Whether employing QL and PC alone or as a QL–PS and PC–PS system, the bioavailable Cd content in the contaminated soil decreased with increasing QL and PC (5–50%). The bioavailable Cd concentration with combined remediation (QL + PS and PC + PS system) was significantly higher than that with ISS amendment alone (QL and PC), supporting that the addition of PS inhibited the Cd immobilization effect of ISS amendment. Under the same amount of amendment (<30%), QL showed a better immobilization efficiency against Cd than PC. Meanwhile, when adding more than 30% QL or PC, the difference in their effects was relatively small. Notably, the hardness of the solidified matrix formed after QL treatment was rather low, and was far lower than that of the solidified matrix formed after PC treatment.

The curing efficiency of aniline varied with the addition of QL and PC (30–50%). The aniline removal rate in co-contaminated soil increased with the addition of QL and PC (Appendix A). The removal rates of aniline were only 76.72% and 57.21% with addition of 50% QL and PC alone, respectively, which did not meet the evaluation criterion for the aniline immobilization rate of more than 90%. The synergy in the QL–PS and PC–PS systems greatly increased the stabilization efficiency of aniline, and the curing rate of aniline was more than 90% when equal or greater than 30% QL and PC was added in the QL–PS and PC–PS systems. Nevertheless, there was no obvious difference with the addition of 50% amendment between the QL–PS and PC–PS systems.

Given that different amounts of QL and PC result in different CaO contents, we speculate that, with increasing CaO content, the curing rate of aniline increases and the bioavailable Cd content decreases, suggesting that CaO has an indispensable role in the immobilization of co-contaminated soil. Moreover, the amount of QL and PC added is an important factor affecting the UCS of the solidified matrix, where greater addition of QL and PC resulted in a higher UCS [41]. However, because compatibilization of QL and PC would appear with increasing addition, it is not advisable to use too large a dosage. Therefore, considering the immobilization rate of aniline and the bioavailable Cd concentration while avoiding compatibilization, we determined 30% to be the optimal dosage of QL and PC in the combined remediation.

### 3.4. Mechanism of Combined In Situ Solidification/Stabilization and Chemical Oxidation

#### 3.4.1. pH and Temperature

The temporal profiles of temperature and pH during the 3-h combined amendment period are shown in Figure 5 and Figure 6, respectively. In the Heat + PS and PS reactions, the temperature remained near the background values of 40 °C and 23 °C throughout the 3-h treatment, respectively. However, all reactions with PS showed a marked temperature increase from the background. As was determined from the contaminant immobilization rates, the maximum temperatures attained in all reactions were directly related to the CaO content of the ISS treatments (Table 1), with temperature increasing as the CaO content decreased. For example, the highest temperature among the ISS treatments (>40 °C) was observed in the QL + PS reaction, followed by PC + PS, BFS + PS, and FA + PS. The reaction of CaO with water (CaO (s) + H_2_O → Ca(OH)_2_ (s), ΔH_298K_ = −104 kJ/mol) releases a large amount of heat. Although it contained no CaO, the Ca(OH)_2_ + PS reaction increased in temperature to nearly 30 °C. As expected, the temperatures increased from background values in all combined remediations, because both alkaline and heat activation cause the rupture of the O−O bond in PS, which releases 140 kJ/mol of energy. Subsequent reactions involving chemical oxidation are also exothermic.

In addition, pH (Figure 6) values helped explain the mechanism of the combined remediation. In the PS reaction, the pH remained approximately stable for the first 60 min, followed by a significant decrease, reaching 7.31 after 3 h of reaction. The pH in the Heat + PS reaction decreased significantly after 15 min of reaction, followed by a slower decrease, reaching 7.22 after 3 h. Interestingly, these results indicate that PS decomposition in heat-activated systems may be superior to that in alkaline-activated systems. The declines in pH during these two reactions were likely driven by the release of SO_4_^2−^ and H^+^ accompanied by the activation of PS, thereby lowering the pH of the soil. With the exception of FA + PS, all activation scenarios reached a pH above 10.5 during the 3-h period, well above the minimum pH required for alkaline PS activation. The highest pH values (>11) were observed in the Ca(OH)_2_ + PS, QL + PS, and PC + PS reactions. Although the pH did not exceed 10.5 in the FA + PS reaction (maximum pH = 10.47), PS activation could be attributed to thermal activation caused by high temperatures (>30 °C). Among the ISS amendments, the maximum pH values increased with CaO content, with the maximum pH observed in the pure CaO treatment (QL); all other ISS amendments contained not only some CaO, but also other constituents that did not influence pH.

Cassidy et al. [27] first reported that adding 50% Ca(OH)_2_ and 50% PC could activate PS in contaminated soil, which demonstrated that ISS amendments could activate PS; however, they failed to distinguish the relative contributions of heat versus alkaline activation. This distinction is important because heat activation generates 2 mol of SO_4_^·−^ per mol of PS, whereas alkaline activation produces only 1 mol of SO_4_^·−^ (S_2_O_8_^2−^ + temperature ≥ 30 °C → 2SO_4_^·−^; S_2_O_8_^2−^ + pH ≥ 10.5 → SO_4_^·−^ + SO_4_^2−^). The activation mechanism of PS in this study followed three regimes: (1) heat activation alone, (2) alkaline activation alone, and (3) a combination of heat and alkaline activation. In the Heat + PS reactor, the reaction temperature exceeded 30 °C, but the pH was below 8, indicating that little or no alkaline activation occurred in this reaction and that heat activation was the main activation mechanism. By contrast, the temperature in the Ca(OH)_2_ + PS reactor did not reach the minimum temperature required for heat activation, but the pH surpassed 10.5, suggesting that alkaline activation was the main activation mechanism. Meanwhile, the combined ISS/ISCO reactors not only had temperatures above 30 °C, but also pH above 10.5 (although the FA + PS treatment pH was close to 10.5). We speculated that some combination of heat and alkaline activation occurs in the combined remediation methods. Meanwhile, it was reasonable to assume that the contribution of heat activation relative to alkaline activation increased as the maximum temperature increased, and with prolonging of the period during which the temperature remained above 30 °C. For instance, the contribution of heat activation relative to alkaline activation would be expected to be greater for QL + PS and PC + PS than for BFS + PS and FA + PS [24].

#### 3.4.2. X-ray Diffraction

Figure 7 shows the XRD patterns of an untreated sample and solidified matrices after seven days of curing. No difference in the characteristic peaks in the patterns were obvious, and the chemical compositions of the solidified matrices were similar. The main SiO_2_ (near 26°) phase and a few products of hydration reactions (e.g., calcium hydroxide, calcium carbonate, ettringite and/or thaumasite, calcium aluminosilicate, and amorphous calcium silicate hydrate (C–S–H)) were identified in the XRD patterns. The presence of a large amount of SiO_2_ in the solidified matrices revealed that active Si was contained in the soil and ISS amendments were triggered, indicative of a trigger driving the curing process. The products of this process connected the surface of the soil and the ISS amendments, and filled the pores of the solidified matrices, integrating the soil into the ISS process to immobilize pollutants. The formation of calcium aluminosilicate and CaCO_3_ may have been due to the reaction of Ca(OH)_2_ and hydrated calcium silicate in the hydration products with carbon dioxide or the secondary reaction of Ca(OH)_2_ and hydrated calcium silicate. Amorphous calcium silicate hydrate appeared only in the ISS treatments alone and was absent from the combined remediation methods, indicating that the addition of PS reduced the adverse effects of aniline hydration. These results indicated that a significant hydration reaction occurred during the combined remediation. Moreover, strong alkalinity and high temperature are the main mechanisms for the rapid removal of pollutants during the oxidative solidification/stabilization process.

#### 3.4.3. Scanning Electron Microscopy

Backscattered SEM images of an untreated sample and solidified matrices after seven days of curing are presented in Figure 8. The particles of the contaminated soil before treatment were fine and dispersed, and those in the solidified structure after seven days of curing were loose, indicating that the presence of aniline adversely affected the compactness of the solidified matrices. The presence of aromatic amines in contaminated soil is an important factor affecting hydration kinetics. The surfaces of the solidified matrices were generally free of cracks and flat. Meanwhile, the exterior of the soil particles was typically coated in hydration products. These results suggest that the concentration of aniline was reduced under the action of chemical degradation of PS, greatly diminishing its effect on the solidified matrix microstructure. Flaky or laminated Ca(OH)_2_ crystals (Figure 8 (c-3,e-3)) and hydrated calcium silicate gel with a net or fibrous structure were observed on the surface of the solidified matrixes (×80,000) (Figure 8 (d-3,f-3)). In addition, small amounts of acicular ettringite (Figure 8 (e-3,f-3)) were observed. However, Ca(OH)_2_ crystals were not observed in the PC + PS reaction. These results indicate that the presence of Ca(OH)_2_ is not conducive to the UCS of the solidified matrices, and the addition of PS has a positive effect on the UCS of the solidified body. The UCS of the PC + PS product was greater than that of the QL + PS product. The net or fibrous structure and crystals mingled with the pores of the solidified matrices coated and coagulated the soil containing the contaminants. Overall, although the UCS of the solidified matrices can be further improved, contaminants can be effectively immobilized inside the solidified matrices, isolating them from environmental media and reducing the risk to the environment.

### 3.5. Evaluation of Remediated Soil

#### 3.5.1. Basic Physical and Chemical Properties

Evaluations of soil remediation technologies should not only consider the contaminant removal effect, but also the impacts on soil quality. The basic physical and chemical properties of the soil before and after the experiment are presented in Table 3. The soil changed from weakly alkaline to strongly alkaline after combined remediation, which was reasonably attributed to the formation of Ca(OH)_2_ via the reaction of CaO with water. This would necessitate the addition of acidic regulators (e.g., calcium superphosphate and dipotassium hydrogen phosphate) to the soil to adjust the pH of the soil. The redox potential of the contaminated soil after the experiment was below −100 mV, indicating that the contaminated soil was in a reduced state and that gas permeability of soil was poor. The pH had a direct influence on the intensity of the ORP. Our results were consistent with the general trend that, in the same soil, higher soil pH was associated with a lower ORP. Soil conductivity reflects the total dissolved salt content of soil and indicates the fertilization capacity of the soil to some extent. The EC of soil changed significantly before and after the experiment, ranging from 3.31 to 121.7 mS/cm and 60.8 mS/cm. The results presented here suggested that the content of total dissolved solids was 779 and 389 mg/kg after combined remediation [42].

The SOM content decreased by 4.24% and 48.31% after the QL + PS and PC + PS reactions, respectively, suggesting that the combined ISS/ISCO treatments were destructive to organic matter, and that PC had a greater influence on SOM than QL. However, there was no significant change in CEC, indicating that the total amount of various cations adsorbed onto soil colloids remained essentially the same, maintaining the original buffer performance. In general, CEC presents a positive correlation with SOM, inconsistent with our results. Wu and Zhou [43] suggested that the relationship between SOM and CEC is very complex, and that the humic acid content of SOM has a strong influence on CEC. Therefore, in the process of combined ISS/ISCO treatment, carbohydrates and nitrogen-containing organic matter with relatively small molecular weights in SOM were greatly affected, while there was minimal effect on humic acid with complex structure and stable properties.

#### 3.5.2. Ca and S Analysis

The addition of ISS amendments (mainly containing CaO) and PS to the soil would cause the accumulation of Ca and S in the soil. Therefore, we examined the Ca and S contents in the soil before and after the experiments (Table 4). The Ca content in the soil increased from 36.11 to 138.90 and 72.92 g/kg after treatment with QL + PS and PC + PS, respectively. Increased soil Ca can promote the coagulation of soil colloids, which is conducive to the formation of agglomerates. Moreover, it can supply Ca required for plant growth. The S content also increased significantly. S in soil mainly exists in solution in the form SO_4_^2−^. Increased soil SO_4_^2−^ confers both positive and negative effects. For instance, soil SO_4_^2−^ can be used by plants as a source of nutrients, thereby promoting growth and production. Furthermore, it also can provide sufficient electron donors and electron acceptors to support the metabolism of microorganisms such as *Thiobacillus* spp. and sulfur-oxidizing bacteria [44]. Nevertheless, SO_4_^2−^ in solution is not easily adsorbed onto soil particles and organic matter, and is readily lost with soil and water, which can increase SO_4_^2−^ concentrations in surface water and groundwater.

#### 3.5.3. Leaching of Toxic Contaminants

Treatment with ISS alone could result in higher leachability of contaminants because organic contaminants can interfere with the cementitious reactions that bind or stabilize contaminants [45]. Table 5 lists the average aniline and Cd concentrations in SPLP extracts from soil samples after seven days of curing. The control reaction did not significantly reduce leachability compared with untreated soil (not shown in Table 5). However, all combined ISS/ISCO treatments achieved the greatest reductions in leaching of aniline. This was because PS activation oxidized a significant portion of aniline, reducing the long-term risk, and was able to reduce the leaching of the contaminants that were not oxidized via ISS treatment. Similar results were obtained in a previous study of soil [46], showing that combined ISS/ISCO treatment performed better than either technology used alone. Meanwhile, combined ISS/ISCO treatment significantly reduced the leaching of Cd, which could meet the national domestic waste landfill pollution control standard (GB 16889-2008).

#### 3.5.4. Unconfined Compressive Strength (UCS) and Hydraulic Conductivity

Table 5 lists the UCS and K values measured in selected treated soil samples. UCS is an important geotechnical property for the final disposal of treated soil, and a common minimum target UCS of 350 kPa at 28 days is suggested by the US EPA guidelines for materials that are to be sent to landfill [47]. In the Netherlands and France, a UCS of 1 MPa is suggested for disposal [48]. Reducing K is an important performance parameter of ISS, because it minimizes the infiltration of rainwater into the treated material, and, in combination with diminished contaminant leachability, minimizes the mobility of contaminants in groundwater. A K value of 1.0 × 10^−9^ m/s is suggested by the US EPA guidelines for disposal.

In this study, The UCS and K of untreated soil were approximately 0.06 MPa and 6.02 × 10^−2^ cm/s after seven days and 0.08 MPa and 2.76 × 10^−2^ cm/s after 28 days, respectively. When the soil samples were cured for seven days, the UCS and K values of single-treatment soil samples were 0.76 MPa and 9.91 × 10^−4^ cm/s for 30% QL and 1.26 MPa and 1.97 × 10^−5^ cm/s for PC. Adding PS had a small impact on the UCS and K values. When the soil samples were cured for 28 days, the UCS and K values rose to 0.91 MPa and 3.07 × 10^−4^ cm/s for the QL treatment and 1.43 MPa and 7.87 × 10^−6^ cm/s for the PC treatment. Adding PS increased the UCS and reduced the K of the treated soil samples in the 30% PC + PS system. These parameters were measured only to determine whether activated PS negatively impacted ISS performance. The above results showed that there was no significant difference in UCS or K values between the ISS/ISCO and ISS treatments, indicating that the reactions associated with activated PS did not negatively impact these two performance parameters.

## 4. Conclusions

We performed batch reactor experiments to assess the efficacy of combined ISS/ISCO treatments in remediating soil contaminated with the organic pollutant aniline and/or the potentially toxic metal Cd. We found that PS could rapidly and effectively remove high concentrations of aniline from contaminated soil. Interestingly, the reaction duration had little effect on the removal efficiency of aniline, although alkaline conditions were beneficial to its degradation. All four ISS amendments tested (QL, PC, BFS, and FA) could activate PS to generate SO_4_^·−^. Moreover, combined ISS/ISCO treatments effectively removed aniline and reduced the bioavailability of Cd. Similarly, the combined ISS/ISCO treatments significantly reduced contaminant leaching. However, PS exhibited an inhibitory effect on the reduction of the bioavailable Cd content, and ISS amendment with PS failed to achieve synergy in the remediation of Cd in co-contaminated soil. The maximum temperature and alkalinity achieved with ISS-activated PS and the percentage of contaminants immobilized both increased with increasing CaO content in the ISS amendments. At the same PS dose, aniline oxidation was enhanced to the extent that heat activation was favored relative to alkaline activation of PS, because the heat mechanism yields two times more oxidizing radicals per mole than the alkaline mechanism of activation. Regardless, both strong alkalinity and high temperature were the main mechanisms for the rapid removal and immobilization of pollutants during the oxidative solidification/stabilization process. The combined treatments had various effects on the basic physical and chemical properties of the soil, and significantly increased the Ca and S contents of the soil. The reactions associated with activated PS did not negatively impact these two ISS performance parameters (UCS and K). Overall, our findings suggest that combined ISS/ISCO treatment has the potential for application to soil co-contaminated with both organic compounds and potentially toxic metals.

In this study, we only performed bench-scale experiments, and pilot-scale tests are warranted for further optimization of the process conditions and to determine the cost before practical application. Furthermore, subsequent molecular level investigations on soils with multiple contaminants should be conducted in the follow-up research. Regardless, the results of this study can be used as a foundation to inform future remediation efforts of soil co-contaminated with both organic pollutants and potentially toxic metals.

## Figures and Tables

**Figure 1 ijerph-15-02595-f001:**
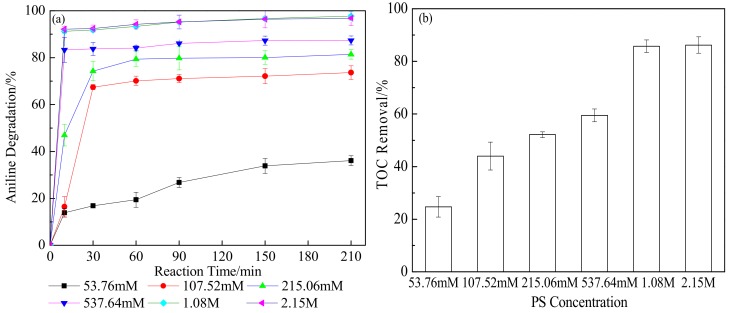
Effects of persulfate (PS) concentration on (**a**) aniline oxidative degradation and (**b**) total organic carbon (TOC) removal (reaction time: 210 min) in soil contaminated with only aniline.

**Figure 2 ijerph-15-02595-f002:**
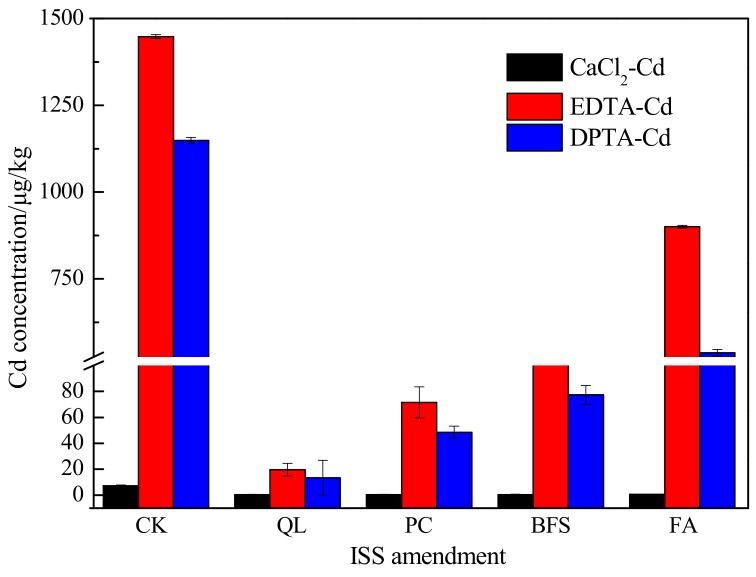
Bioavailable content of Cd in soil contaminated with only Cd. No amendments added (CK), Quick lime (QL), Portland cement (PC), blast-furnace slag (BFS), and fly ash (FA) treatment: CaCl_2_-Cd content <0.5 μg/kg.

**Figure 3 ijerph-15-02595-f003:**
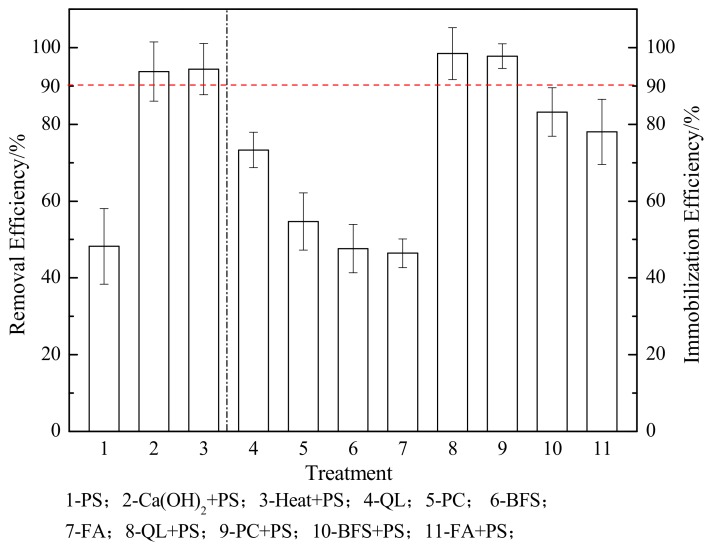
Effect of different treatments on aniline removal and immobilization efficiency in soil co-contaminated with aniline and Cd. Persulfate (PS), Quick lime (QL), Portland cement (PC), blast-furnace slag (BFS), and fly ash (FA).

**Figure 4 ijerph-15-02595-f004:**
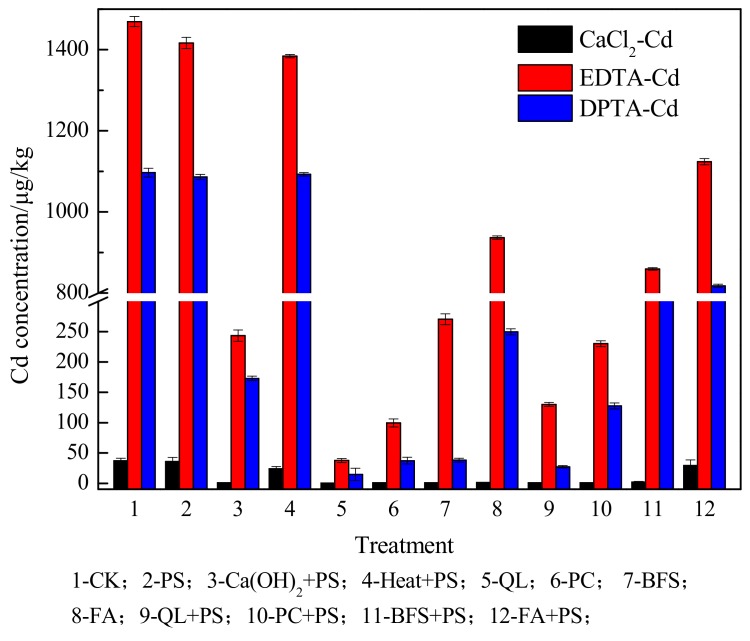
Effect of different treatments on the bioavailable Cd content in soil co-contaminated with aniline and Cd. See Appendix A for the treatment abbreviations. No amendments added (CK); Persulfate (PS); Quick lime (QL), Portland cement (PC), blast-furnace slag (BFS), and fly ash (FA).

**Figure 5 ijerph-15-02595-f005:**
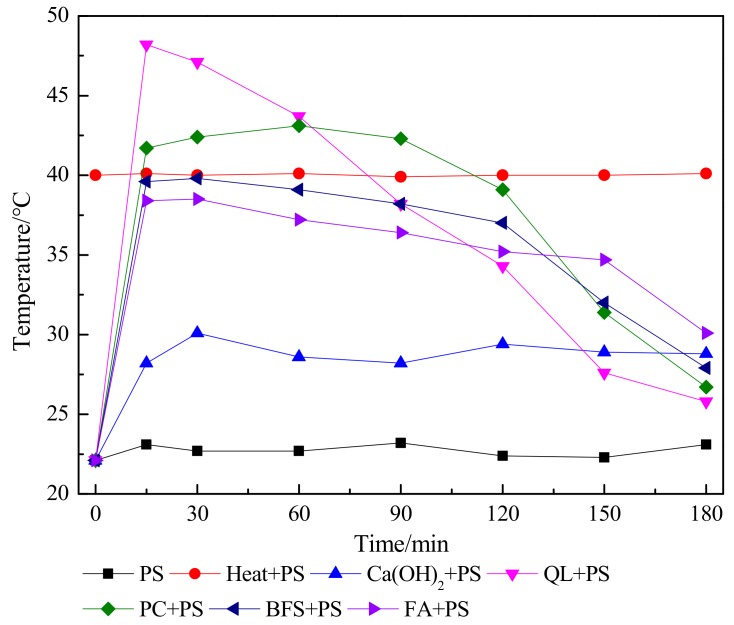
Temperature measurements during the 3-h mixing period in all reactions amended with persulfate. See Appendix A for the treatment abbreviations. Persulfate (PS); Quick lime (QL), Portland cement (PC), blast-furnace slag (BFS), and fly ash (FA).

**Figure 6 ijerph-15-02595-f006:**
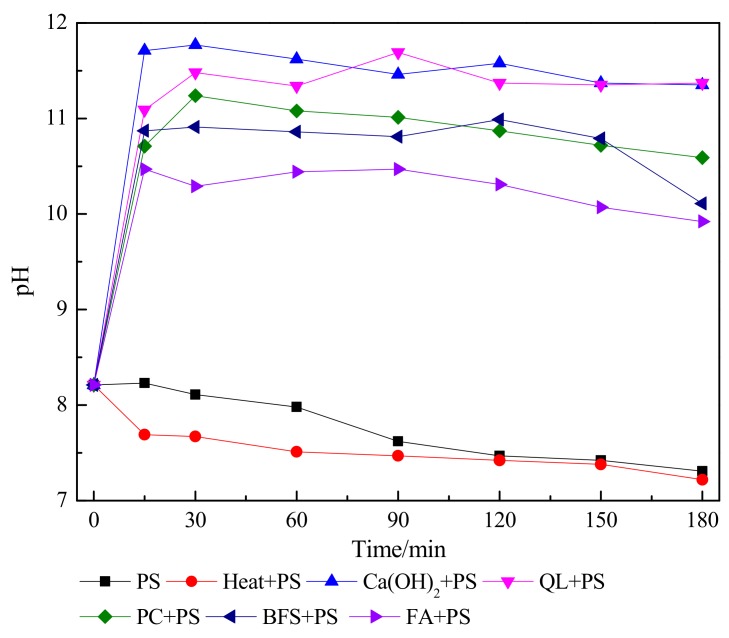
The pH of the system during the 3-h mixing period in all reactions amended with persulfate. See Appendix A for the treatment abbreviations. Persulfate (PS), Quick lime (QL), Portland cement (PC), blast-furnace slag (BFS), and fly ash (FA).

**Figure 7 ijerph-15-02595-f007:**
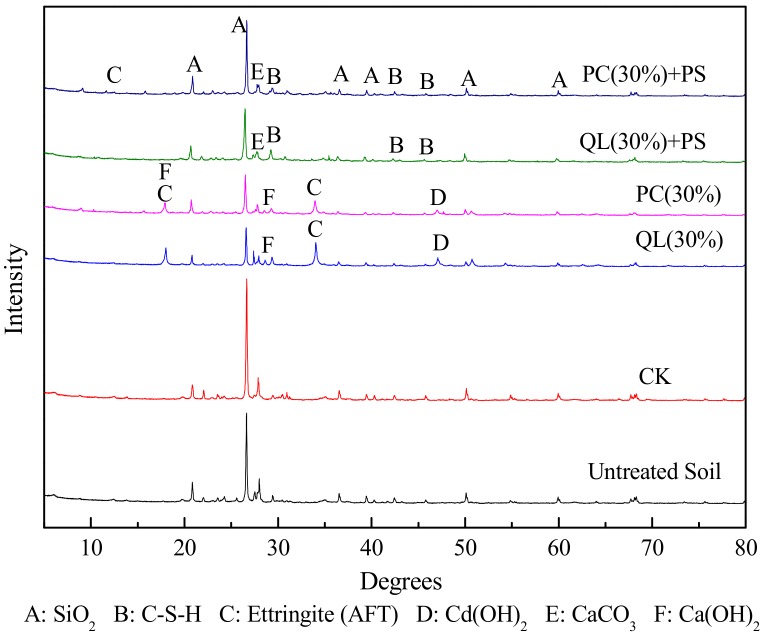
X-ray diffraction patterns of an untreated sample and solidified matrixes after seven days of curing. No amendments added (CK), Persulfate (PS), Quick lime (QL), Portland cement (PC).

**Figure 8 ijerph-15-02595-f008:**
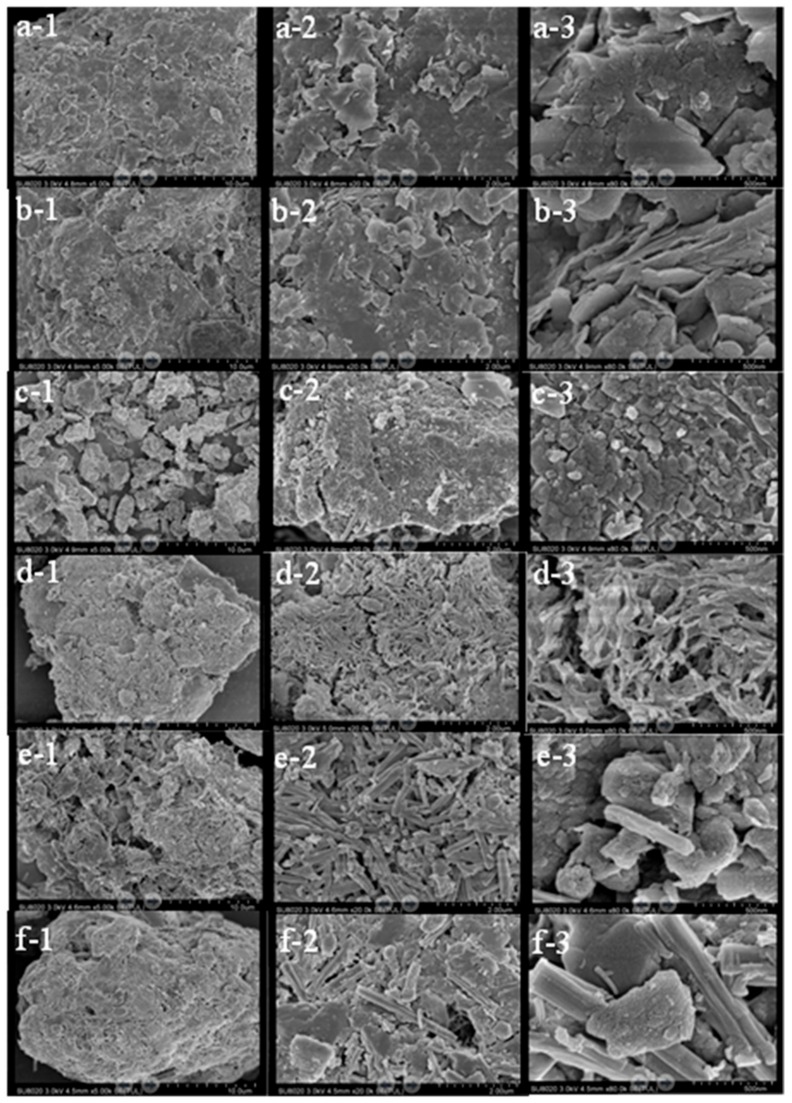
Scanning electron microscopy micrographs of an untreated sample and solidified matrices after seven days of curing: (**a**) untreated, (**b**) CK, (**c**) QL (30%), (**d**) QL (30%) + PS, (**e**) PC (30%), (**f**) PC (30%) + PS; (**1**) ×5,000, (**2**) ×20,000, (**3**) ×80,000. No amendments added (CK); Persulfate (PS); Quick lime (QL), Portland cement (PC).

**Table 1 ijerph-15-02595-t001:** The four in situ solidification/stabilization (ISS) amendments tested, their abbreviations, and the CaO contents. QL—quick lime; PC—Portland cement; BFS—blast-furnace slag; FA—fly ash.

ISS Amendment	Abbreviation	CaO Content (%) [24,30,31]
Quick lime (CaO)	QL	100%
Portland cement	PC	60–68%
Blast-furnace slag	BFS	40–45%
Fly ash	FA	21–27%

**Table 2 ijerph-15-02595-t002:** Bioavailable fraction of heavy metals in soil [37]. EDTA—ethylenediaminetetraacetic acid; DPTA—diethylenetriaminepentaacetic acid; TEA—triethanolamine.

Form	Reagent (Concentration, mol/L)	Water: Soil Ratio (*v*/*m*)	Oscillation Time (min)
CaCl_2_-Cd	CaCl_2_ (0.01)	1:5	120
EDTA-Cd	EDTA (0.05)	1:5	30
DPTA-Cd	Mixed liquor containing DPTA (0.005), CaCl_2_ (0.01), and TEA (0.1)	2:5	120

**Table 3 ijerph-15-02595-t003:** Basic physical and chemical properties of the soil before and after reaction.

Treatment	pH	ORP (mV)	EC (mS/cm)	SOM (%)	CEC (cmol/kg)
Untreated	8.22	−85.9 ± 0.26 ^a^	3.31 ± 0.92 ^a^	1.18 ± 0.43 ^a^	8.25 ± 1.72 ^a^
30% QL + PS	12.44	−342.7 ± 0.42 ^b^	121.7 ± 0.85 ^b^	1.13 ± 0.32 ^a^	8.02 ± 0.94 ^a^
30% PC + PS	11.07	−265.1 ± 0.94 ^c^	60.8 ± 1.72 ^c^	0.61 ± 0.86 ^b^	7.81 ± 0.98 ^a^

ORP: oxidation–reduction potential; EC: electrical conductivity; SOM: soil organic matter; CEC: cation exchange capacity; QL: quick lime; PS: persulfate; PC: Portland cement. The data are presented as the mean value ± standard deviation (*n* = 3). Values denoted with different letters in each column differ significantly (*p* < 0.05).

**Table 4 ijerph-15-02595-t004:** Content of Ca and S before and after the experiments.

Treatment	Ca content (g/kg)	S content (g/kg)
Untreated	36.11 ^a^	0.28 ^a^
30% QL + PS	138.90 ^b^	8.81 ^b^
30% PC + PS	72.92 ^c^	10.66 ^c^

PC: Portland cement; PS: persulfate; QL: quick lime. Values denoted with different letters in each column differ significantly (*p* < 0.05).

**Table 5 ijerph-15-02595-t005:** Leaching of toxic contaminants and the unconfined compressive strength (UCS) and hydraulic conductivity (K) of treated soil samples cured for 7 or 28 days.

Treatment	Concentration of Leached Aniline (mg/kg)	Concentration of Leached Cd (mg/kg)	UCS (MPa)		K (cm/s)	
			7 Days	28 Days	7 Days	28 Days
CK	162.87	1.72	0.06	0.08	6.02 × 10^−2^	2.76 × 10^−2^
30% QL	-	-	0.76	0.91	9.91 × 10^−4^	3.07 × 10^−4^
30% QL + PS	2.77	0.13	0.74	0.84	8.14 × 10^−4^	2.95 × 10^−4^
30% PC	-	-	1.26	1.43	1.97 × 10^−5^	7.87 × 10^−6^
30% PC + PS	6.42	0.25	1.36	1.67	2.02 × 10^−5^	5.88 × 10^−6^

CK: control; PC: Portland cement; PS: persulfate; QL: quick lime.

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
