# Peer review of "Remediating Potentially Toxic Metal and Organic Co-Contamination of Soil by Combining In Situ Solidification/Stabilization and Chemical Oxidation: Efficacy, Mechanism, and Evaluation"

_ijerph, 2018, doi:10.3390/ijerph15112595_

Round 1
Reviewer 1 Report
1. General Comments:
This manuscript investigated the "remediating heavy metal and organic contamination in soil by combining in situ solidification/stabilization and chemical oxidation"
The in situ technologies presented involve applying chemical, biological, or physical processes to the subsurface to degrade, remove, or immobilize contaminants without removing the bulk soil. Compared to excavation and ex situ treatment, the use of these technologies offers several benefits, such as addressing deep contamination and generally costing less (EPA, 2006).
EPA (2006). In Situ Treatment Technologies for Contaminated Soil. EPA 542/F-06/013
2. Abstract:
This part was arranged well.
3. Introduction:
Adequate.
4. Materials and Methods:
Aniline oxidation by PS
Support this part with previous studies. Support methods, ranges, dosages and etc.
5. Results and Discussion:
The discussion part should be improved. You should compare your results with previous studies in details.
Author Response
#Reviewer 1
1. General Comments:
This manuscript investigated the "remediating heavy metal and organic contamination in soil by combining in situ solidification/stabilization and chemical oxidation"
The in situ technologies presented involve applying chemical, biological, or physical processes to the subsurface to degrade, remove, or immobilize contaminants without removing the bulk soil. Compared to excavation and ex situ treatment, the use of these technologies offers several benefits, such as addressing deep contamination and generally costing less (EPA, 2006).
EPA (2006). In Situ Treatment Technologies for Contaminated Soil. EPA 542/F-06/013
Response: We are grateful for your time spent reviewing our manuscript and for your comments, which have helped us to improve the quality of our research and manuscript.
2. Abstract:
This part was arranged well.
3. Introduction:
Adequate.
4. Materials and Methods:
Aniline oxidation by PS
Support this part with previous studies. Support methods, ranges, dosages and etc.
Response: We appreciate your careful review and academic rigor. The selected methods and dosages for aniline oxidation by persulfate are based on previous studies, which we have added to the manuscript (Xie et al., 2012; Wu et al., 2015) (Page 4, Lines 139, 144–145, 150–152).
References:
Xie, X.; Zhang, Y.; Huang, W.; Huang, S. Degradation kinetics and mechanism of aniline by heat-assisted persulfate oxidation. J. Environ. Sci. China. 2012, 24(5), 821-826.
Wu, Y. Study on the rapid disposal of the aniline contaminated soil by solidification/stabilization technology. Ocean University of China. 2015. (in Chinese)
5. Results and Discussion:
The discussion part should be improved. You should compare your results with previous studies in details.
Response: Thank you for your comments. Following your suggestion, we have compared our results with those of previous studies to help contextualize our findings (Page 5, Lines 196–197, 202–204, 208–213; Page 6, Lines 216–217, 221–223; 225-232; Page 7, Lines 268–270, 279; Page 8, Lines 280–281; 289-293; 301-304; Page 9, Lines 322–324; Page 10, Line 375; Page 11, Lines 408–412; Page 15, Lines 511–512, 516–518).

Reviewer 2 Report
Remediating commingled heavy metal and organic 2 contamination in soil by combining in situ 3 solidification/stabilization and chemical oxidation: 4 efficacy, mechanism, and evaluation. Yan Ma , Zhenhai Liu , Yanqiu Xu , Shengkun Zhou , Yi Wu , Jin Wang , Zhanbin Huang , Yi Shi,
REVIEW
This manuscript should be published after minor revision. The authors make a valid point about the remediation of soils with multiple contaminants. It is an engineering type of research, and could lead to subsequent molecular level research on soils with multiple contaminants.
This initial phase of the research should be followed by a molecular level investigation of the kinetics and mechanisms. That would support quantitative predictions for the planning of remediation projects. The next phase of the research would need molecular level information about stoichiometries, chemical reactions, and related physical processes. Stoichiometry would require the use of chemical units for amounts and concentrations.
A “Future Research” section at the end of the manuscript should indicate the need for this next part of th research. The time dependent curves in Figure 1 show that a followup project could investigate the molecular level kinetics of the mechanisms.
Following this first manuscript, the next manuscript would report the basic science that is the necessary base for the important technology. When the technology has been developed from the basic science, a third manuscript could report it.
Author Response
# Reviewer 2
This manuscript should be published after minor revision. The authors make a valid point about the remediation of soils with multiple contaminants. It is an engineering type of research, and could lead to subsequent molecular level research on soils with multiple contaminants.
This initial phase of the research should be followed by a molecular level investigation of the kinetics and mechanisms. That would support quantitative predictions for the planning of remediation projects. The next phase of the research would need molecular level information about stoichiometries, chemical reactions, and related physical processes. Stoichiometry would require the use of chemical units for amounts and concentrations.
A “Future Research” section at the end of the manuscript should indicate the need for this next part of the research. The time dependent curves in Figure 1 show that a followup project could investigate the molecular level kinetics of the mechanisms.
Following this first manuscript, the next manuscript would report the basic science that is the necessary base for the important technology. When the technology has been developed from the basic science, a third manuscript could report it.
Response: Thank you for your careful review and comments, in particular your perspective on the remediation of soils with multiple contaminants, which have helped us improve our paper and guide our research.
Based on existing research, we carried out a series of experiments on combined heavy metal and organic contamination in soil. This paper represents the first part of a longer-term systematic research plan with the main goal of developing an efficient technique for the remediation of high-concentration co-contaminated soil. According to your suggestion, we plan to conduct follow-up research. To reflect this, we have added a brief discussion of future research directions in section 4 (Page 16, Lines 572–573).

Reviewer 3 Report
The article (ijerph-388002) “Remediating commingled heavy metal and organic contamination in soil by combining in situ solidification/stabilization and chemical oxidation: efficacy, mechanism, and evaluation” provides a number of significant results that may be interesting for the scientific community. However, I think it needs a major revision before publication. The discussion of the results appears very poor. Therefore, without this clarification, it is difficult for me to recommend the manuscript for publication in its present form in ijerph.
Title and in the manuscript Line 68 “heavy metals” not is appropriate, they are call potential toxic metals/elements (i.e. https://doi.org/10.3390/ijerph15030543)
Introduction
Page 2 Line 54 need a reference (i.e. physicochemical remediation: doi:10.3390/su10030636 and bioremediation: https://doi.org/10.3390/app8081336)
Page 2 Line 56 need a reference (i.e. doi: 10.1007/s11356-016-6804-0)
Section 2
Insert a section with material
Line 114 motivate the choice 0-20 cm?
Line 115 motivate the choice 0.5 cm?
Line 119 The authors need to provide more detailed information about the choice of these procedure or insert the standard methods
Line 120 motivate the choice of the contamination at these concentrations
Line 137 The authors need to provide more detailed information about the choice of these procedure or insert the standard methods
Line 145 Add the statistical analysis in the methodology and in discussion (i.e. ANOVA)
Line 151 The authors need to provide more detailed information about the choice of these procedure or insert the standard methods
Line 155 The authors need to provide more detailed information about the choice of these procedure or insert the standard methods
Section 3.1 The results are well present, however the discussion is poor and the authors need to improve it.
Section 3.3.1 The results are well present, however the discussion is poor and the authors need to improve it.
Author Response
# Reviewer 3
The article (ijerph-388002) “Remediating commingled heavy metal and organic contamination in soil by combining in situ solidification/stabilization and chemical oxidation: efficacy, mechanism, and evaluation” provides a number of significant results that may be interesting for the scientific community. However, I think it needs a major revision before publication. The discussion of the results appears very poor. Therefore, without this clarification, it is difficult for me to recommend the manuscript for publication in its present form in ijerph.
Response: We appreciate your academically rigorous and careful review. We have considered your comments carefully and made corrections accordingly, as outlined below.
1. Title and in the manuscript Line 68 “heavy metals” not is appropriate, they are call potential toxic metals/elements (i.e. https://doi.org/10.3390/ijerph15030543)
Response: We are grateful for your comment. Following your suggestion, we have revised this term throughout the manuscript to “potentially toxic metal” (Page 1, Lines 2, 19, 36, 42; Page 2, Lines 47, 48, 54, 57; Page 3, Line 104; Page 6, Line 235; Page 9, Line 324; Page 16, Lines 550, 569, 575).
2. Page 2 Line 54 need a reference (i.e. physicochemical remediation: doi:10.3390/su10030636 and bioremediation: https://doi.org/10.3390/app8081336)
3. Page 2 Line 56 need a reference (i.e. doi: 10.1007/s11356-016-6804-0)
Response: Thank you for your comments and we appreciate your carefulness. Following your suggestion, we have cited these references in the introduction (Page 2, Lines 55, 57; Page 17, Lines 612–616, 629–631).
4. Insert a section with material
Response: Thank you for your comment and we apologize for the confusing organization of this section. The list of materials is included in section 2.1 (Chemicals and materials). To clarify this, we have divided this information into additional sub-sections 2.1.1 (Chemicals), 2.1.2 (Test soil sampling and processing), and 2.1.3 (ISS amendments) (Page 3, Lines 108, 116, 131).
5. Line 114 motivate the choice 0-20 cm?
Response: We appreciate your careful review. The test soil used in the experiment was artificially contaminated. The soil was collected from farmland in Tongzhou District, Beijing, China, from the 0–20-cm soil layer (i.e., cultivated soil layer), which experiences the most obvious physical, chemical, and biological effects within the soil horizon.
6. Line 115 motivate the choice 0.5 cm?
Response: We are grateful for your comment. The soil was sieved to remove particles larger than 0.5 cm in diameter mainly to remove stones and plant debris (Srivastava et al., 2016.). We have added this explanation and reference to the manuscript (Page 3, lines 118–119).
Reference:
Srivastava, V.J.; Hudson, J.M.; Cassidy, D.P. Achieving synergy between chemical oxidation and stabilization in a contaminated soil. Chemosphere. 2016, 154, 590-598.
7. Line 119 The authors need to provide more detailed information about the choice of these procedure or insert the standard methods
Response: We appreciate your comment and academic rigor. Following your suggestion, we have provided more detailed information on the preparation of aniline-contaminated soil (Page 3, Lines 121–125).
8. Line 120 motivate the choice of the contamination at these concentrations
Response: Thank you for your comment. According to previous reports (Tang et al., 2010; Yuan et al., 2010), high-concentration pollution is common in soil co-contaminated with potentially toxic metal(s) and organic pollutants (e.g., the concentrations of metals and organic pollutants in the soil near the sewage pipe of a chemical plant were 14.28 mg/kg and 4850 mg/kg, respectively). As such, our main goal is to ultimately develop an efficient remediation technique for high-concentration co-contamination of soil. We have added this explanation and these references to explain the choice of pollutant concentrations used in this study (Page 3, Lines 128–130).
References:
Tang, X.; Shen, C.; Shi, D.; Cheema, S.A.; Khan, M.I.; Zhang, C.; Chen, Y. Heavy metal and persistent organic compound contamination in soil from Wenling: An emerging e-waste recycling city in Taizhou area, China. J. Hazard. Mater. 2010, 173(1-3), 653-660.
Yuan, S.; Wu, X.; Wan, J. Enhanced washing of HCB and Zn from aged sediments by TX-100 and EDTA mixed solutions. Geoderma. 2010, 156, 119-125.
9. Line 137 The authors need to provide more detailed information about the choice of these procedure or insert the standard methods
Response: We appreciate your comment and academic rigor. Following your suggestion, we have provided detailed information on aniline oxidation with persulfate (Page 4, Lines 139; 144-145; 150-152).
10. Line 145 Add the statistical analysis in the methodology and in discussion (i.e. ANOVA)
Response: Thank you for your comment. Following your suggestion, we have added information on the statistical analyses to sections 3.5.1 and 3.5.2 (Pages 14–15, Table 3, Table 4).
11. Line 151 The authors need to provide more detailed information about the choice of these procedure or insert the standard methods
12. Line 155 The authors need to provide more detailed information about the choice of these procedure or insert the standard methods
Response: We appreciate your comment and academic rigor. Following your suggestion, we have provided detailed information on persulfate oxidation and soil solidification/stabilization (Page 4, Lines 158–161, 163–168).
13. Section 3.1 The results are well present, however the discussion is poor and the authors need to improve it.
Response: Thank you for your comment. Following your suggestion, we have supplemented and refined the discussion in this section, including comparisons with previous studies and analysis of possible explanations for our findings (Page 5, Lines 196–197, 202–204, 208–213; Page 6, Lines 221-223; 225–232).
14. Section 3.3.1 The results are well present, however the discussion is poor and the authors need to improve it.
Response: We appreciate your comment. Following your suggestion, we have supplemented and refined the discussion in this section (Page 7, Lines 268–270, 279; Page 8, Lines 280-281; 289-293; 301–304; Page 9, Lines 322-324).

Round 2
Reviewer 3 Report
This revised version has improved greatly. However not all critical point are resolved.
5. Line 114 motivate the choice 0-20 cm?
Response: We appreciate your careful review. The test soil used in the experiment was artificially contaminated. The soil was collected from farmland in Tongzhou District, Beijing, China, from the 0–20-cm soil layer (i.e., cultivated soil layer), which experiences the most obvious physical, chemical, and biological effects within the soil horizon
Need to insert the standard method
Author Response
Reviewer #3: This revised version has improved greatly. However not all critical point are resolved.
5. Line 114 motivate the choice 0-20 cm?
Response: We appreciate your careful review. The test soil used in the experiment was artificially contaminated. The soil was collected from farmland in Tongzhou District, Beijing, China, from the 0–20-cm soil layer (i.e., cultivated soil layer), which experiences the most obvious physical, chemical, and biological effects within the soil horizon.
The standard method:
We appreciate your careful review. Soil drilling was used as a means to collect soil. The surface debris at the sampling point was removed before the soil was collected, and the drill was inserted vertically into the soil to a depth of 0-20 cm (marked on the drill with a scale). We have included the method in the revised manuscript (Page 8, Lines 118-120). Thank you sincerely once again.
